# Infective Endocarditis after Transcatheter Aortic Valve Replacement: Challenges in the Diagnosis and Management

**DOI:** 10.3390/pathogens12020255

**Published:** 2023-02-05

**Authors:** Johnny Zakhour, Fatima Allaw, Suha Kalash, Saliba Wehbe, Souha S. Kanj

**Affiliations:** 1Department of Internal Medicine, Division of Infectious Diseases, American University of Beirut Medical Center, Beirut P.O. Box 11-0236, Lebanon; 2Center for Infectious Diseases Research, American University of Beirut, Beirut P.O. Box 11-0236, Lebanon

**Keywords:** transcatheter aortic valve replacement, infective endocarditis, prosthetic valve endocarditis

## Abstract

Although initially conceived for high-risk patients who are ineligible for surgical aortic valve replacement (SAVR), transcatheter aortic valve replacement (TAVR) is now recommended in a wider spectrum of indications, including among young patients. However, similar to SAVR, TAVR is also associated with a risk of infectious complications, namely, prosthetic valve endocarditis (PVE). As the number of performed TAVR procedures increases, and despite the low incidence of PVE post-TAVR, clinicians should be familiar with its associated risk factors and clinical presentation. Whereas the diagnosis of native valve endocarditis can be achieved straightforwardly by applying the modified Duke criteria, the diagnosis of PVE is more challenging given its atypical symptoms, the lower sensitivity of the criteria involved, and the low diagnostic yield of conventional echocardiography. Delay in proper management can be associated with increased morbidity and mortality. Therefore, clinicians should have a high index of suspicion and initiate proper work-up according to the severity of the illness, the underlying host, and the local epidemiology of the causative organisms. The most common causative pathogens are Gram-positive bacteria such as *Staphylococcus aureus,* coagulase-negative staphylococci, *Enterococcus* spp., and *Streptococcus* spp. (particularly the viridans group), while less-likely causative pathogens include Gram-negative and fungal pathogens. The high prevalence of antimicrobial resistance complicates the choice of therapy. There remain controversies regarding the optimal management strategies including indications for surgical interventions. Surgical assessment is recommended early in the course of illness and surgical intervention should be considered in selected patients. As in other PVE, the duration of therapy depends on the isolated pathogen, the host, and the clinical response. Since TAVR is a relatively new procedure, the outcome of TAVR-PVE is yet to be fully understood.

## 1. Introduction

Transcatheter aortic valve replacement (TAVR) is a minimally invasive procedure that was initially developed for the management of symptomatic severe aortic valve stenosis. The technique was first performed in 2002 on an inoperable 57-year-old patient who had previously undergone balloon aortic valvuloplasty [1]. Since then, TAVR has gained increased interest and established a sizeable clinical and market value, with more than 110,000 procedures performed from 2015 through 2017 in the United States (US) alone, and a global market worth 2 billion USD dollars annually [2,3]. With TAVR being much less invasive than surgical aortic valve replacement (SAVR) and due to the accumulating evidence of its favorable outcomes, starting in 2016, the food and drug administration (FDA) gradually expanded its indications to include low- and intermediate-risk patients [4].

In view of the expanding number of TAVR procedures, rare complications including infective endocarditis have been reported. Complications may manifest as catastrophic aortic root dissection, paravalvular and aortic root abscesses, and intra/paravalvular regurgitation [5]. Studies have estimated that the 1-year incidence of prosthetic valve endocarditis (PVE) complicating TAVR (TAVR-PVE) ranges between 0.2% (in the PARTNER 3 trial) and 3.1% (in a Danish cohort including 509 patients) [4,6]. Additionally, the partner 1A trial and the NOTION trial recorded a cumulative 5-year incidence of 2% and 6.2%, respectively [7,8]. Although uncommon, TAVR-PVE is associated with a high in-hospital mortality rate, which reached 34.4% in one systematic review [9]. In a pooled analysis of patients from the PARTNER 1 and PARTNER 2 trials, TAVR-PVE was found to be an independent risk factor for mortality [10]. However, this risk is not higher compared to SAVR. A meta-analysis of four comparative randomized controlled trials (RCT) that analyzed around 3800 patients during a mean follow-up period of 3.4 years found no significant difference in the frequency of endocarditis between both approaches. Furthermore, a meta-analysis that included over 84,000 patients concluded that the short- and long-term risk of PVE appears to be identical in patients undergoing TAVR and SAVR regardless of the type of valve used, the duration of follow-up, or the patients’ surgical risk [11].

The diagnosis of TAVR-PVE may be challenging due to the low sensitivity of conventional diagnostic tools and the panoply of non-specific symptoms [5]. In light of its high mortality and dreadful complications, clinicians should have a low threshold to consider TAVR-PVE in patients who present with suggestive symptoms and signs and should initiate prompt management. It should be noted that mortality remains elevated in patients who were only managed with antibiotics as well as those who also underwent repeat surgery [9].

Although years of experience with SAVR-PVE have generated a considerable amount of evidence and number of recommendations, this may not be the case for TAVR patients with PVE, who may present with different demographics, symptoms, and causative organisms. Hence, clinicians from different specialties should be familiar with the clinical presentation, diagnosis, and microbiology of TAVR-PVE in order to optimize the care of these patients and avoid detrimental complications. Herein, we shall elucidate the disease’s predictors and risk factors and discuss the diagnostic challenges and pertinent microbiological data while focusing on the differences between TAVR and SAVR. Finally, we will review the latest evidence on prevention strategies and the antimicrobial and surgical management of TAVR-PVE.

## 2. Risk Factors and Predictors of TAVR-PVE

### 2.1. Patient-Related Risk Factors

Younger age and male sex have been the most commonly reported non-modifiable risk factors for TAVR-PVE [9,12,13]. In fact, a recent meta-analysis including over 68,000 patients reported that older age was associated with a significantly lower risk of TAVR-PVE (RR 0.97, 95% CI: 0.95–0.99, and *p* = 0.007) [13]. One suggested explanation is that younger patients requiring TAVR have more significant comorbidities compared to their older counterparts. For instance, a prospective study including 1448 patients who underwent TAVR found that younger patients with TAVR-PVE had significantly higher rates of diabetes, chronic kidney disease (CKD) requiring hemodialysis (HD), prior cardiac surgery, and chronic obstructive pulmonary disease (COPD) [14]. The lower incidence of TAVR-PVE among the female sex has been attributed to the protective effects of estrogen towards the vascular endothelium; however, additional studies are warranted to thoroughly investigate the underlying mechanisms [12,14]. Although most studies report male sex as a risk factor for TAVR-PVE, a recent retrospective cohort study revealed that the incidence was similar between both genders. Moreover, female sex was identified as an independent risk factor for mortality [15].

Modifiable risk factors that affect the patients’ prognosis should be identified and addressed, if possible, as they may have significant implications in terms of management [10]. The reported patient-related risk factors for PVE include diabetes mellitus (41.7% vs. 30.0%; HR, 1.52; 95% CI, 1.02–2.29) [16], peripheral arterial disease (PAD), COPD, CKD, and HD [17]. Moreover, a retrospective analysis of three combined national registries from Sweden found that body surface area, critical pre-operative state, and atrial fibrillation were significant patient-related predictors of TAVR-PVE (Table 1) [18].

### 2.2. Procedural Risk Factors

In addition to patient-related risk factors, some procedural factors have been implicated in the predisposition to TAVR-PVE. A meta-analysis and systematic review including 57,531 patients found that oro-tracheal intubation (RR 2.99, 95% CI:2.73–3.28, and *p* < 0.001) was a significant risk factor for TAVR-PVE [18]. In fact, a study showed that over 10% of oro- or naso-tracheal intubations are complicated by *Staphylococcus* spp. or *Streptococcus* spp. bacteremia [19]. Furthermore, TAVR has been associated with an increased incidence of high-degree atrio-ventricular (AV) block requiring pacemaker implantation, which has been shown to increase the risk of TAVR-PVE (RR 5.19, 95% CI: 4.16–6.47, and *p* < 0.001) [18].

The impact of the valve’s type, whether a self-expanding valve (SEV) or balloon expendable valve (BEV), on the incidence and mortality of TAVR-PVE is debatable. A recent meta-analysis including 30 studies and 73,780 patients found a numerically higher incidence of PVE among patients with BEV valves compared to those with SEV (0.8% vs. 0.3%); however, these findings were not significant [12]. Additionally, Tinica et al. reported that SEV may be associated with lower mortality in TAVR-PVE (OR 0.39, 95% CI 0.16 to 0.98, and *p* < 0.05) [20]. On the other hand, a subgroup analysis in a study including 1895 patients undergoing TAVR found no significant difference in the rates of TAVR-PVE among patients with SEV or BEV [21]. Similarly, data from the multicenter “Infectious Endocarditis After TAVR International Registry” including 6398 patients reported that neither the incidence of PVE nor the in-hospital mortality (34.4% in the SEV group vs. 37.4% in the BEV group; *p*  =  0.63) were correlated with the type of valve [16].

The route of vascular access (trans-femoral vs. non-trans-femoral) does not seem to impact the incidence of TAVR-PVE, as most studies have not been able to establish a definitive association [12]. For instance, a meta-analysis including over 1200 patients reported no association between trans-femoral vascular access and the risk of TAVR-PVE (RR 0.85, 95% CI: 0.71 to 1.02, and *p* = 0.08) [13]. Similar results were reported from a pooled analysis of all the patients from the PARTNER 1 and PARTNER 2 trials (*p* = 0.8) [10].

Residual aortic regurgitation (AR) is a common finding that has been reported in the majority of patients following TAVR [22]. It may predispose patients to PVE either through mechanical damage caused by the high-velocity regurgitate jet or through thrombus formation that serves as a nidus for infection [23,24]. Early incidence of TAVR-PVE was significantly associated with post-procedural AR, particularly moderate to severe regurgitation (22.4% vs. 14.7%; HR, 2.05; 95% CI, 1.28–3.28) [16,18]. This association could have implications for the management of patients post-TAVR, as those with moderate to severe AR likely require prophylactic antibiotics before undergoing procedures associated with a high risk of bacteremia.

### 2.3. Duration since TAVR

The timing of TAVR-PVE relative to the procedure may be classified as very early (within 30 days), early (within 30–60 days), intermediate (between 60 and 365 days), or late (after 365 days [25]. Similar to endocarditis following SAVR, most cases of TAVR-PVE occur within the first year post-implantation followed by an exponentially decreasing incidence over time [26]. One registry including 250 definite cases of TAVR-PVE reported that the median time from TAVR to infective endocarditis was 5.3 months (interquartile range [IQR], 1.5–13.4 months) [16]. Another national registry from Sweden reported the incidence to be 1.42% during the first year, 0.80% during the following 1–5 years, and 0.52% at 5–10 years following the procedure [18]. The higher incidence of PVE during the early post-procedural period may be explained by the presence of risk factors for bacteremia (indwelling catheters, central lines, and critical illness), nosocomial infections, and the incomplete endothelialization of the newly implanted valve [18,27].

## 3. Diagnostic Challenges of TAVR-PVE

Patients with TAVR-PVE may present with vague complaints and non-specific symptoms, which may delay the diagnosis and appropriate management [28]. In fact, acute heart failure is one of the most common presentations of TAVR-PVE, alongside vague, nonspecific symptoms such as malaise and fatigue [25]. One study that included 7891 patients with TAVR-PVE found that heart failure was the second most common presenting symptom (58.5%) after fever (71.7%) [9]. Although the modified Duke criteria are highly sensitive for the diagnosis of native valve endocarditis (NVE), they are not as sensitive regarding the diagnosis of PVE [24]. The lack of sensitivity may reflect on the epidemiological data of TAVR-PVE, as the use of the Duke criteria may underestimate its real incidence [24]. In one study comparing the Duke criteria to those of the European Society of Cardiology (ESC), which include multimodal imaging, ESC’s modified criteria were found to have a sensitivity of 100% compared to only 50% for Duke’s criteria [29].

Echocardiography is the first-line imaging modality for diagnosing IE [29]. Although trans-esophageal echocardiography (TEE) is more sensitive than trans-thoracic echocardiography (TTE), TTE is more often performed due to its lower cost and convenience [30]. However, in PVE, particularly TAVR-PVE, TTE may not be the optimal tool to establish a diagnosis [31]. In fact, the high content of metal in TAVR valves and the distribution of metal struts within the valve may impair the visualization of vegetations, especially smaller ones [25]. Additionally, vegetations are three times more likely to be detected on SEVs than BEVs during echocardiography because of the wider frame [16]. Interestingly, one report by Mangner et al. showed that vegetations were observed in only 25% of 47 patients with TAVR-PVE while normal results were reported in up to 31.9%, confirming that TTE has a low diagnostic yield [17]. TEE may also be unable to differentiate between vegetations and fibrous strands or thrombi [32]. Hence, more advanced modalities of imaging are required, such as fluorodeoxyglucose-positron emission tomography (FDG-PET) and single-proton emission CT scans, to diagnose TAVR-PVE (Figure 1) [32]. However, despite being integrated in the ESC’s guidelines, such advanced imaging techniques remain inaccessible and insufficiently performed, especially in low-resource settings [33]. The addition of an abnormal FDG uptake during FDG-PET as a major criterion for the diagnosis of PVE may improve the diagnostic yield. For example, in a prospective study including patients with suspected PVE, the addition of FDG-PET to the modified Duke criteria as a major criterion significantly increased its sensitivity from 70% to 98% (*p*-value = 0.008) [34]. It should be noted that persistent vegetations after the completion of the treatment course are often not an indicator of treatment failure and should not incite the continuation of therapy beyond 6 weeks if clinical, biological, and microbiological cures are documented. However, it is advised that patients with persistent vegetations undergo repeat echocardiography as the increase in vegetation size or a size greater than 10 mm have been associated with poorer outcomes [35].

In light of all the diagnostic challenges of TAVR-PVE, it is encouraged that a multidisciplinary team consisting of cardiologists, infectious diseases specialists, microbiologists, cardiac imaging specialists, and cardiothoracic surgeons be involved in the care of the patient [36]. Delays in establishing the diagnosis should be avoided to reduce mortality and spare the need for surgical intervention in this subset of patients who are at a high operative risk [37,38].

## 4. Microbiology of TAVR-PVE

Like all cases of endocarditis, identifying the causative organism is essential for the successful targeted management of TAVR-PVE. However, in the setting of severe presentation and critical illness, select patients may require empiric antimicrobial therapy prior to the identification of a causative organism [20]. In fact, the causative organism may not be isolated in up to 12% of TAVR-PVE, and in such cases, empiric therapy is the cornerstone of treatment [39]. Knowledge of the local epidemiology and microbiological profile of infections is essential to guide empiric therapy. This is particularly important for multi-drug-resistant (MDR) pathogens, which may complicate the course of treatment and increase mortality. Hence, medical centers are encouraged to identify their local epidemiological data to guide empirical therapy in accordance with antimicrobial stewardship principles.

### 4.1. Bacterial TAVR-PVE

In most reports of TAVR-PVE, Gram-positive organisms seem to be predominant [28]. In a recently published systematic review and meta-analysis, *Streptococcus* spp. (25.3%), *Staphylococcus* spp. (25.3%), and *Enterococcus* spp. (24.1%) were the most commonly reported causative organisms of TAVR-PVE (Table 2) [20]. Among staphylococci and enterococci, *S. aureus* and *E. faecalis* were the predominant species (60% and 65.8%, respectively). Interestingly, enterococci have been much more commonly reported in TAVR-PVE compared to SAVR-PVE, possibly owing to the trans-femoral access of TAVR [40]. Moreover, enterococci seem to be more common in TAVR-PVE in patients who have SEVs rather than BEVs [41].

Although Gram-negative pathogens are rarely reported as causative organisms for TAVR-PVE, they are often responsible for other nosocomial infections after TAVR in hospitalized patients. For instance, a recent prospective cohort study of 303 patients reported that 17% of patients developed nosocomial infections following TAVR. In those patients, Gram-negative organisms were isolated in more than 60% of cultures [46]. Additionally, an urgent indication of TAVR, the length of coronary care unit (CCU) stays, and the need for blood transfusion were identified as independent risk factors for post-TAVR nosocomial infections via multivariate analysis [46]. Since most of these infections were pneumonias and urinary tract infections, the risk for secondary bacteremia is non-negligeable; hence, such patients may be at increased risk for Gram-negative PVE. Gram-negative infections should also be considered in TAVR-PVE occurring early after TAVR, as a meta-analysis suggests that the median time from TAVR to PVE was 1.1 months (IQR 0.9–6.2) for Gram-negative pathogens and 5.4 months (IQR 0.9–6.2) for Gram-positive bacteria [20]. The prevalence of Gram-negative organisms among causative organisms of TAVR-PVE is very low. Thus, most large studies do not report the genus of Gram-negative organisms and combine them into one group of patients. Among those reporting the genus of Gram-negative organisms, the results are highly variable and thus conclusions cannot be drawn as to which Gram-negative organisms are the most common.

Besides the increased incidence of enterococcal infection in the early phase after TAVR, and the association of early TAVR-PVE with hospital-acquired *S. aureus,* there are no notable differences in the microbiology of early (<1 year) and late (>1 year) TAVR-PVE [43]. However, since up to 52% of TAVR-PVE may be hospital-acquired, the possibility of the presence of nosocomial MDR organisms should be considered in cases of early TAVR-PVE [16]. A recently published multicenter study including 91 patients with very early TAVR-PVE (<30 days after TAVR) reported that up to one-third of the isolated *Staphylococcus* spp. (35.2%) were methicillin-resistant, while the second most common organism was *Enterococcus* spp. (34.1%) [47]. These rates are comparable to those published among patients with endocarditis occurring after 30 days of TAVR. It should be noted that in up to 50% of TAVR-PVE cases, the primary source of bacteremia cannot be identified [20]. However, if identified, a primary site of infection causing secondary bacteremia and IE can help guide empiric therapy by targeting the most likely causative organisms, particularly when blood cultures are negative.

### 4.2. Other Causative Organisms of TAVR-PVE

Fungal endocarditis in patients with a history of TAVR is rather rare but highly deadly [47]. The reported rates of fungal TAVR-PVE range between 0.8% and 3% of all TAVR-PVE cases [9,16,43]. Due to its low incidence, the true mortality of fungal TAVR-PVE is not well known. When compared with bacterial etiologies, it has been associated with an increased risk of mortality (aHR: 1.72; 95% CI: 1.23–2.39; *p* < 0.05) [43]. Besides case reports of TAVR-PVE caused by *Candida parapsilosis* [45,48] and *Histoplasma capsulatum* [49,50], the microbiology of these fungal infections is hard to characterize given that most large studies do not report the genus nor species of causative fungal organisms. Fungal cases of TAVR-PVE are challenging to treat given the high production of biofilm, the difficulty of eradication, the risk of relapse, and the need for surgical intervention [51].

Although multiple case series of PVE caused by *Mycobacterium* spp. have been reported in the literature, they almost exclusively involve patients who had undergone SAVR whose infection was thought to be due to the contamination of operating theaters [52,53,54]. Although the evidence on the microbiology of TAVR-PVE is rapidly accumulating, to date, there are no reports of TAVR-PVE caused by *Mycobacterium* spp. The involvement of mycobacteria in TAVR-PVE would have great implications for the management of these infections as they are often difficult to eradicate and require combination and prolonged therapy.

### 4.3. Blood Culture-Negative TAVR-PVE

Blood culture-negative endocarditis (BCNE) can greatly complicate the management of patients with endocarditis. Some studies have even reported rates as high as 70% [55]. Notably, patients with BCNE may require a prolonged duration of broad-spectrum antimicrobial therapy, which places them at risk for adverse events such as drug toxicity and *Clostridioides difficile* infection. Most commonly, BCNE is due to exposure to antimicrobials prior to blood cultures or to endocarditis caused by fastidious organisms [39]. Novel diagnostic techniques other than standard blood cultures may help identify more causative organisms and offer more targeted antimicrobial therapy. Molecular sequencing techniques such as metagenomic next-generation sequencing or targeted metagenomic sequencing and 16s rRNA PCR have been shown to improve the diagnostic yield in infective endocarditis, including PVE [56]. Such advanced technologies can also be applied to patients with culture-negative TAVR-PVE.

## 5. Infection Prevention and Control

### 5.1. Peri-Procedural Prophylaxis

The risk of TAVR PVE seems to be the highest during the first year following the procedure, which may be due to the incomplete healing and occasional persistence of paravalvular leaks for up to 12 months after the procedure [27]. Peri-procedural bacteremia may also contribute to this early increased risk [27]. Hence, patients undergoing TAVR may benefit from peri-procedural antimicrobial prophylaxis [57]. There are no recommendations that are based on robust evidence regarding the choice of antimicrobial agents for pre-procedural prophylaxis. According to a recent systematic review and meta-analysis, most patients receive peri-procedural prophylaxis with cephalosporins (61.8%), while others receive vancomycin (16%) or penicillin (22%) [9]. Although cefazolin is the most commonly used agent, it exhibits no activity against *Enterococcus* spp. and hence may not be the most appropriate agent for prophylaxis [58]. The rationale behind the use of first or second-generation cephalosporins was extrapolated from data on SAVR [59]. Nevertheless, compared to SAVR, the likelihood of the involvement of enterococci is much higher [38]. Hence, prophylactic regimens for use during cardiac surgery may not be adequate for TAVR. A subset of patients may also require intracardiac devices for conduction anomalies (e.g., pacemakers), which could further alter the microbiological profile of these infections [60]. Patients admitted for TAVR often have multiple comorbidities, which results in frequent contact with healthcare and colonization with MDR pathogens [16]. Although some studies have reported that Gram-negative organisms might be more frequently encountered during the early period following TAVR-PVE, their incidence is too small and unlikely to have implications on the choice of antimicrobial agent for preoperative prophylaxis [58]. Eventually, prophylaxis should be guided by each center’s epidemiological data and rates of resistance, most importantly the rates of MRSA and vancomycin resistant enterococci (VRE).

Regarding the choice of antimicrobial prophylaxis, a single dose of amoxicillin–clavulanate (2.2 g intravenous 0–60 min prior to arterial puncture) or ampicillin–sulbactam (3 g intravenous 0–60 min prior to arterial puncture) is more appropriate than cephalosporins, wherein another dose is administered if the procedure lasts for more than 2 h [59]. In cases of beta-lactam allergy or high rates of MRSA, a single dose of IV vancomycin (15 mg/kg to be infused over one to 2 h) or teicoplanin (9–12 mg/kg) may be administered. If local rates of VRE are high, teicoplanin (9–12 mg/kg) or daptomycin (>10 mg/kg) given as a single dose are suitable alternatives [61]. It should be noted, however, that teicoplanin is not available in the US. The combination of vancomycin with a first- or second-generation cephalosporin may offer synergistic activity in settings with very high rates of MRSA [62]. In patients with positive MRSA screening or diabetes with a BMI > 30 kg/m^2^, decolonization with a chlorhexidine shower bath and nasal mupirocin ointment is recommended 1–6 weeks before the procedure [59]. On the other hand, patients with frequent hospitalizations and risk factors for VRE colonization should be screened to optimize prophylactic regimens [63]. In fact, a recently published prospective cohort including 290 patients who were admitted for trans-femoral vascular interventions (angiography, TAVR, and foramen closure) reported that enterococci were isolated in 16.6% of patients prior to disinfection and 1.4% after disinfection [64]. Moreover, special attention should be paid to skin care, oral and dental hygiene, dialysis catheters and other invasive catheters, and the optimization of serum glucose levels. There is little evidence regarding the benefits of a longer duration of periprocedural antimicrobial prophylaxis. For instance, a retrospective study comparing a 1-day prophylaxis regimen with cefuroxime to a 3-day regimen in 450 patients undergoing TAVR showed no advantage of the longer regimen. A longer duration of prophylaxis was associated with a significantly higher incidence of *C. difficile* colitis (4% vs. 0.4% *p* = 0.01) [65]. Given the lack of evidence concerning the benefits of longer prophylactic regimens, antimicrobial prophylaxis administered as a single dose (or two doses if the procedure is prolonged) is more appropriate.

### 5.2. Infection Control and Prevention during TAVR

Infection control and prevention measures should be implemented during TAVR procedures to minimize the risk of nosocomial infections, which are often caused by MDR pathogens that further complicate the management of TAVR-PVE. Unlike SAVR, which is performed in the operating room (OR) where strict, evidence-based infection control measures are implemented, TAVR might be performed in a catheterization lab or a hybrid OR where strict infection control measures are not applied. However, a recent study showed no difference in the rates of PVE in patients who underwent TAVR in a hybrid OR compared to those who underwent the procedure in a catheterization laboratory [16]. Nevertheless, the infection control and prevention measures in a diagnostic catheterization laboratory may be insufficient for an implanted prosthetic device [59]. The restriction of traffic inside the room, use of surgical hand hygiene, sterile gowns, gloves and masks, pre-procedural chlorhexidine showers, and unpacking the valve immediately prior to implantation are all examples of practices that may help minimize the risk of infection, despite the lack of evidence [59]. Although there are no data comparing chlorhexidine and iodine in terms of vascular access sterilization in TAVR, data may be extrapolated from studies showing superiority of chlorhexidine at preventing post-procedural BSI in central vascular lines or surgical procedures [66,67].

### 5.3. Post-Procedural Prophylaxis

Despite the absence of randomized trials assessing the benefits of prophylaxis in this subset of patients, the American Heart Association (AHA) and the ESC recommend antimicrobial prophylaxis before dental interventions involving the gingiva, genitourinary and gastrointestinal procedures, and TEE for high-risk patients including those with a history of TAVR [33]. Most recently, a scientific statement from the AHA published in 2021 reconfirmed that antimicrobial prophylaxis is warranted in patients with TAVR following dental procedures, alongside meticulous oral hygiene and routine dental care [68]. However, this may be debatable, with reports confirming greater adverse events from antimicrobial prophylaxis prior to such procedures rather than benefits. In fact, it is estimated that the risk of bacteremia is higher during routine daily activities such as tooth brushing and picking than during dental visits; hence, antibiotic prophylaxis would avoid few, if any, episodes of endocarditis [69]. If prophylaxis is chosen, the agent of choice is oral amoxicillin given as a single dose of 2 g, but alternative oral (cephalexin, azithromycin, clarithromycin, and doxycycline) and intravenous (ampicillin, cefazolin, and ceftriaxone) agents are acceptable for patients who are allergic to penicillin and/or cannot tolerate oral intake [68]. Patients should be educated regarding their increased risk of endocarditis and the importance of avoiding activities that increase the risk of bacteremia such as piercings and tattoos and intravenous (IV) drug use, and they should be instructed to maintain proper dental hygiene.

## 6. Management Challenges

### Surgical Intervention

The dogma concerning various infections is that source control, particularly in terms of sources involving foreign bodies, is highly advisable as it can drastically reduce the pathogen inoculum, eradicate the source of infection, and substantially reduce mortality [70]. Source control in TAVR-PVE requires surgical intervention and advantages should be carefully weighed against surgical risks. In fact, many patients who have undergone TAVR have multiple comorbidities and contraindications that put them at a high operative risk [16]. Most commonly, surgical interventions include the retrieval of the large prosthetic frame adherent to the annulus and ascending aorta, valve replacement, and, in some cases, the replacement of the aortic root, and may prove to be very challenging [9].

Once TAVR-PVE is diagnosed, the cardiac surgery team should be consulted. However, there is no consensus regarding the absolute need for surgical intervention for the management of TAVR-PVE, as most evidence is derived from observational studies and no randomized trials have compared surgical management to conservative management with antimicrobials alone. In light of the absence of guidelines that are specific to TAVR-PVE, extrapolating from the guidelines on the management of PVE may be appropriate at this time while more conclusive evidence emerges. As stated earlier, a multidisciplinary approach involving all stakeholders is warranted alongside a meticulous surgical risk assessment. The 2015 AHA guidelines recommend surgery for all cases of PVE [71]. On the other hand, the ESC 2015 guidelines regarding the management of infective endocarditis recommend surgical intervention for PVE cases only when there are concerns regarding cardiogenic shock of a valvular origin, uncontrolled infection (persistent bacteremia, abscesses, enlarging vegetation, fistula, or false aneurysm), fungal or resistant bacterial organisms, and embolic threats (large vegetations) [33].

The first registry reporting outcomes of surgical interventions versus conservative management for TAVR-PVE included 205 patients. Surgery was only performed in 14.8% of patients despite the fact that 81.2% of them had at least one indication for surgery according to the aforementioned guidelines. Most importantly, surgery was not associated with reduced in-hospital death (29.7% vs. 37.1%, *p* = 0.39). However, these findings may have been caused by selection bias [16]. Similarly, an unmatched cohort comparing outcomes between 20 patients with TAVR who underwent surgical management and 44 patients who were managed conservatively also did not find a significant difference in mortality among both groups. When the subjects were matched, all-cause 1-year mortality was also not significantly different among both groups (65% in patients treated surgically vs. 75% in patients treated with antibiotics alone; *p* = 0.49) [72]. Additionally, another multicenter study including 584 patients with TAVR-PVE of which 19% underwent surgery and 81% were managed with antimicrobials alone did not find a significant difference in hospital and 1-year all-cause mortality. Notably, surgery was less likely to be performed for older patients and those with neurologic symptoms, and was more likely to be performed for those with large vegetations >10 mm, peri-annular complications, systemic embolization, and persistent bacteremia [73]. In other reports, patients who effectively underwent surgery were also more likely to be young, have a lower baseline Surgical Thoracic Surgeons (STS) score reflecting a lower operative risk, and a higher glomerular filtration rate [74]. Given that most observational studies mention selection bias regarding surgical management and did not report surgical outcomes in specific subpopulations (e.g., patients with uncontrolled infection, fungal or *S. aureus* infection), randomized trials are needed to further elucidate the role of surgical intervention in the management of TAVR-PVE.

## 7. Antimicrobial Treatment

### 7.1. Empirical Antimicrobial Therapy of TAVR-PVE

The course of infective endocarditis can be insidious. In hemodynamically stable patients with a subacute illness, antimicrobial therapy can be deferred until every attempt is made to identify the causative organism and susceptibility results [39]. However, in cases of severe illness such as septic or cardiogenic shock, empirical therapy should be initiated, preferably after three sets of blood cultures are drawn from different sites (if possible) [33]. The AHA’s guidelines, which have been endorsed by the Infectious Diseases Society of America (IDSA), recommend that infectious diseases specialists should be consulted to provide guidance regarding the proper empirical regimen [39], which should cover staphylococci, enterococci, and streptococci, as these are the most commonly encountered pathogens in TAVR-PVE. The spectrum should be guided by local epidemiological data and rates of resistance. Additionally, when selecting antimicrobial agents, risk factors for MDR pathogens (intensive care unit (ICU) admission, critical illness, indwelling catheters, central lines, and recent broad-spectrum antimicrobial therapy) and the possibility of concomitant bacterial and fungal infections should be considered in certain scenarios.

A combination of bactericidal antibiotics including aminoglycosides is recommended. The rationale behind using aminoglycosides lies in their synergistic effect when combined with cell-wall inhibitors (B-lactams and glycopeptides), which optimizes their bactericidal activity [33]. The choice of coverage of MRSA during empirical antimicrobial therapy should be guided by the local epidemiological data and rates of community-acquired and hospital-acquired MRSA. It is reasonable that coverage of MRSA be initiated empirically with vancomycin. Antibiotic de-escalation once identification and susceptibility results are available is highly advised. The choice of empirical vancomycin is also appropriate given the high prevalence of enterococci among the causative organisms of TAVR-PVE. The addition of rifampicin is not warranted for empirical therapy. Although it is characterized by its ability to penetrate adherent prostheses and biofilms and exhibits a potent bactericidal effect against *S. aureus* organisms, rifampicin should not be administered until 3–5 days after effective antimicrobial therapy when the bacterial load has decreased and the risk of treatment-emergent resistance is lower [33,75,76]. On the other hand, *Pseudomonas aeruginosa* is rarely a causative organism of infective endocarditis [77]. However, empirical antipseudomonal therapy should be initiated in critically ill patients with risk factors for *P. aeruginosa* infection [78]. A combination of a beta-lactam agent such as ceftazidime, cefepime, or piperacillin-tazobactam in addition to vancomycin and an aminoglycoside (gentamicin or amikacin) is appropriate. However, if local rates of MDR *P. aeruginosa* are high, or if the patient is known to be colonized with MDR organisms, a carbapenem or a novel beta-lactam–beta-lactamase inhibitor combination (ceftolozane–tazobactam, or ceftazidime–avibactam) should be considered. It should be reiterated that there are no randomized trials comparing different antimicrobial agents for TAVR-PVE, and recommendations are extrapolated from studies wherein the majority of patients have undergone prosthetic SAVR.

### 7.2. Directed Antimicrobial Therapy of TAVR-PVE

Directed therapy for TAVR-PVE is similar to SAVR-PVE and should be guided by the susceptibility profile of the causative organism as per the international guidelines [33,39]. There are no randomized controlled trials that have performed a head-to-head comparison of various antimicrobial agents specifically in TAVR-PVE. Agents with good biofilm activity are preferred.

As mentioned previously, the epidemiology of fungal TAVR-PVE is unclear given the rarity of this event. Hence, antimicrobial management is highly variable depending on the causative fungus. Based on the recommendations of the AHA guidelines regarding PVE, a combined medical and surgical approach is recommended for the management of TAVR-PVE caused by *Candida* spp. [39]. Nonetheless, the guidelines stem from conflicting evidence regarding the impact of surgical intervention on mortality [79,80]. Moreover, no current data have indicated the superiority of any antifungal agent compared to another. Hence, the use of a lipid formulation of amphotericin B (daily administration of three to five mg/kg intravenously) with or without flucytosine (25 mg/kg orally administered four times daily) or echinocandins at a high dose (caspofungin 150 mg IV daily, micafungin 150 mg IV daily, and anidulafungin 200 mg IV daily), as advised by the AHA guidelines, is acceptable [51]. Speciation and susceptibility testing of all fungal isolates should be performed to determine the appropriateness of treatment and allow for potential step-down therapy if possible. Even if surgery is performed, antifungal therapy should be continued for at least 6 weeks. Treatment duration may be extended according to clinical and laboratory parameters as the risk of relapse is high in candida endocarditis. If surgery is not performed, lifelong suppression with an oral azole (according to the isolate susceptibilities) may be warranted [51].

### 7.3. Duration of Therapy

Adequate duration of therapy is among the most important pillars of antimicrobial stewardship. As the evidence accumulates against long durations of treatment, most guidelines have moved towards recommending shorter duration of therapy for various infections (particularly Gram-negative pathogens) [81]. However, little to no data are available regarding the efficacy of a shorter duration of treatment for PVE. The AHA and ESC’s guidelines both recommend a minimum duration of antibiotics of 6 weeks for PVE compared to NVE. In fact, prolonged treatment is needed for all types of PVE, as biofilms tend to be formed on the implanted valve and bacteria may remain dormant in the vegetations and display tolerance towards most antimicrobials [33]. According to the guidelines, patients with SAVR-PVE who have undergone surgery that yielded positive cultures from valvular material should receive a post-operative antibiotic course of six weeks while a shorter course of two weeks is warranted among those with negative intra-operative cultures [33]. These recommendations regarding the duration of treatment according to the intra-operative cultures are valid regardless of any delay in surgical intervention. When surgery is delayed, there is no clear guidance on duration of therapy post-operatively according to valvular culture positivity. We believe that the duration of therapy should depend on the clinical response, follow-up inflammatory markers, and tolerance of the prescribed regimen.

### 7.4. Route of Administration

It is recommended that all patients with endocarditis (including those with NVE and PVE) receive initial parenteral treatment. In fact, parenteral administration may provide a more predictable serum concentration than oral drugs and enhance bacterial killing [82]. However, some studies have attempted transitioning to oral therapy in alignment with antimicrobial stewardship principles. A recent RCT including 400 patients with endocarditis, of which 26% had prosthetic valves, found that transitioning to oral therapy after a minimum of 10 days of parenteral therapy was not inferior to completing the treatment course parenterally [83]. The data from this study should be cautiously extrapolated to TAVR-PVE since the population size was small and heterogenous and none of the patients had severe illness. Due to the lack of evidence regarding the safety of transitioning to oral therapy for TAVR-PVE, we believe that a full-course parenteral antimicrobial treatment is recommended to optimize patients’ outcomes.

## 8. Conclusions

TAVR-PVE has emerged in light of the increasing number of patients undergoing TAVR. Contrarily to SAVR-PVE, clinicians have less experience identifying, diagnosing, and managing TAVR-PVE. Studies have shown a similar rate of incidence between PVE among patients with SAVR and TAVR but have identified different risk factors. Notably, younger age and male sex are the most important non-modifiable risk factors. Additionally, other co-morbidities and procedural risk factors may play a role in TAVR-PVE. Most importantly, TAVR-PVE tends to have a different microbiological profile than SAVR-PVE. The overwhelming prevalence of enterococci in TAVR-PVE has major implications regarding empiric therapy, peri-procedural and post-procedural prophylaxis, and infection prevention and control measures. As little evidence exists regarding the management of these infections, clinicians have no choice but to extrapolate from previous guidelines on the management of SAVR-PVE. The decision for surgical intervention and the choice of empirical antimicrobial therapy should be personalized and adjusted to the local epidemiological data and rates of resistance. Although most clinicians have promoted transitioning to shorter duration of therapy or to oral therapy, there is not enough evidence to recommend such practices in TAVR-PVE.

## Data Availability

Not applicable.

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
