# Peer review of "Infective Endocarditis after Transcatheter Aortic Valve Replacement: Challenges in the Diagnosis and Management"

_pathogens, 2023, doi:10.3390/pathogens12020255_

Round 1

Reviewer 1 Report

The author did an excellent job reviewing this complex topic with extensive literature support. These are minor comments to consider for clarification and potentially strengthening this paper.  

1.      Line 148, the reference is missing. It showed as “(ref).”

2.      Line 155, “Patients with TAVR-PVE do not often present with the typical signs and symptoms of endocarditis, like fever or new onset heart murmur.” However, the author also mentioned that a previous study has shown that the most common symptom was fever (71.7%). These two sentences contradicted each other.   

3.      For Scheme 1, it would be helpful to add a box adjacent to the “Suspected TAVR-PVE” box with the risk factor or clinical presentations leading to the clinical suspicion for PVE. Moreover, the author should clarify the “ESC 15 criteria” and “Initiate management Re-evaluate routinely.” I do not clearly understand.

4.      Line 237, it would be helpful to mention the genus of gram-negative bacteria that has a higher tendency to cause TAVR-PVE than others, e.g., HACEK group, ?Pseudomonas, ?Serratia.

5.      It may be worthwhile to touch on other unusual organisms such as Mycobacterium: as there was an M. chimera outbreak of PVE in the past.

6.      Line 287, “Gram-negative organisms may be more frequent during the early period following TAVR-PVE.” I am not sure that the incident of GNB is high enough (compared to the GPC) to alter the preoperative prophylaxis.  

7.      The antimicrobial prophylaxis pre-TAVR at our institution is based on the same protocols used for cardiac surgery: cefazolin. However, the point is that not all institutions use the same prophylaxis strategy. Furthermore, the evidence may not be strong enough to conclude that ampicillin-sulbactam is more appropriate than cefazolin.

8.      The duration of perioperative prophylaxis needs further clarification (line 315). There is no clear evidence for antibiotic > 48 hours in any cardiac surgery procedure. Therefore, we used only two doses of cefazolin at our institution.

9.      It may be worth noting that teicoplanin is not available in the US.  

10.  The recent scientific statement: Prevention of Viridans Group Streptococcal Infective Endocarditis (Circulation 2021), should be discussed in the dental prophylaxis section.

11.  Regarding the duration of therapy, it would be interesting to discuss the duration of antibiotics for some common scenarios. For example, some patients did not undergo surgery at the time of diagnosis, yet they underwent delayed surgery after completing 6 weeks of antibiotics. Should the antibiotic course be restarted after this delayed surgery, if the intraoperative culture (or pathology) were positive?   

12.  Some references were not in the correct format. For example, references 1 and 2 were in the internet search format rather than the typical bibliography.  

Reviewer 2 Report

I congratulate the authors for such an interesting narrative review on this contemporary theme.

I have some comments/concerns.

1.       Please confirm the reference for the paragraph starting on line 142 (mentioned reference 25) on page 3/17. I failed to see the written information in that reference.

2.       Line 147 on the same page – a reference is missing.

3.       Line 253 on page 7/17 – regarding fungal IE in TAVR-PVE. No specific reference to TABR-PVE is provided in the reference 54.

4.       Reference re-check with the manuscript should be performed by the authors.

5.       The subchapter on risk factors is too dense – I propose that a table is inserted for easier reading. Again, effort should be made to improve the manuscript's overall aspect (use of Tables, for example, using Text formatting, etc)
